# Information Geometry Theoretic Measures for Characterizing Neural Information Processing from Simulated EEG Signals

**DOI:** 10.3390/e26030213

**Published:** 2024-02-28

**Authors:** Jia-Chen Hua, Eun-jin Kim, Fei He

**Affiliations:** 1Centre for Fluid and Complex Systems, Coventry University, Coventry CV1 2NL, UK; ejk92122@gmail.com; 2Centre for Computational Science and Mathematical Modelling, Coventry University, Coventry CV1 2TL, UK; fei.he@coventry.ac.uk

**Keywords:** information geometry, information length, information rate, causal information rate, causality, stochastic oscillators, electroencephalography, stochastic simulation, signal processing, dementia, Alzheimer’s disease, information theory, neural information processing, brain networks

## Abstract

In this work, we explore information geometry theoretic measures for characterizing neural information processing from EEG signals simulated by stochastic nonlinear coupled oscillator models for both healthy subjects and Alzheimer’s disease (AD) patients with both eyes-closed and eyes-open conditions. In particular, we employ information rates to quantify the time evolution of probability density functions of simulated EEG signals, and employ causal information rates to quantify one signal’s instantaneous influence on another signal’s information rate. These two measures help us find significant and interesting distinctions between healthy subjects and AD patients when they open or close their eyes. These distinctions may be further related to differences in neural information processing activities of the corresponding brain regions, and to differences in connectivities among these brain regions. Our results show that information rate and causal information rate are superior to their more traditional or established information-theoretic counterparts, i.e., differential entropy and transfer entropy, respectively. Since these novel, information geometry theoretic measures can be applied to experimental EEG signals in a model-free manner, and they are capable of quantifying non-stationary time-varying effects, nonlinearity, and non-Gaussian stochasticity presented in real-world EEG signals, we believe that they can form an important and powerful tool-set for both understanding neural information processing in the brain and the diagnosis of neurological disorders, such as Alzheimer’s disease as presented in this work.

## 1. Introduction

Identifying quantitative features from neurophysiological signals such as electroencephalography (EEG) is critical for understanding neural information processing in the brain and the diagnosis of neurological disorders such as dementia. Many such features have been proposed and employed to analyze neurological signals, which not only resulted in insightful understanding of the brain neurological dynamics of patients with certain neurological disorders versus healthy control (CTL) groups, but also helped build mathematical models that replicate the neurological signal with these quantitative features [1,2,3,4,5].

An important distinction, or non-stationary time-varying effects of the neurological dynamics, is the switching between eyes-open (EO) and eyes-closed (EC) states, where numerous research studies have been conducted on this distinction between EO and EC states to quantify important features of CTL subjects and patients using different techniques on EEG data such as traditional frequency-domain analysis [6,7], transfer entropy [8], energy landscape analysis [9], and nonlinear manifold learning for functional connectivity analysis [10], while also attempting to relate these features to specific clinical conditions and/or physiological variables, including skin conductance levels [11,12], cerebral blood flow [13], brain network connectivity [14,15,16], brain activities in different regions [17], and performance on the unipedal stance test (UPST) [18]. Clinical physiological studies found that there are distinct mental states related to the EO and EC states. Specifically, there is an “exteroceptive” mental activity state characterized by attention and ocular motor activity during EO, and an “interoceptive” mental activity state characterized by imagination and multisensory activity during EC [19,20]. Ref. [21] suggested that the topological organization of human brain networks dynamically switches corresponding to the information processing modes when the brain is visually connected to or disconnected from the external environment. However, patients with Alzheimer’s disease (AD) show loss of brain responsiveness to environmental stimuli [22,23], which might be due to impaired or loss of connectivities in the brain networks. This suggests that dynamical changes between EO and EC might represent an ideal paradigm to investigate the effect of AD pathophysiology and could be developed as biomarkers for diagnosis purposes. However, sensible quantification of robust features of these dynamical changes between EO and EC of both healthy and AD subjects, solely relying on EEG signals, is nontrivial. Despite the success of many statistical and quantitative measures being applied to neurological signal analysis, the main challenges stem from the non-stationary time-varying dynamics of the human brain with nonlinearity and non-Gaussian stochasticity, which makes most, if not all, of these traditional quantitative measures inadequate, and blindly applying these traditional measures to nonlinear and nonstationary time series/signals may produce spurious results, leading to incorrect interpretation.

In this work, by using simulated EEG signals of both CTL groups and AD patients under both EC and EO conditions and based on our previous works on information geometry [24,25,26], we develop novel and powerful quantitative measures in terms of information rate and causal information rate to quantify the important features of neurological dynamics of brains. We are able to find significant and interesting distinctions between CTL subjects and AD patients when they switch between the eyes-open and eyes-closed status. These quantified distinctions may be further related to differences in neural information processing activities of the corresponding brain regions, and to differences in connectivities among these brain regions, and therefore, they can be further developed as important biomarkers to diagnose neurological disorders, including but not limited to Alzheimer’s disease. It should be noted that these novel and powerful quantitative measures in terms of information rate and causal information rate can be applied to experimental EEG signals in a model-free manner, and they are capable of quantifying non-stationary time-varying effects, nonlinearity, and non-Gaussian stochasticity presented in real-world EEG signals, and hence, they are more robust and reliable than other information-theoretic measures applied to neurological signal analysis in the literature [27,28]. Therefore, we believe that these information geometry theoretic measures can form an important and powerful tool set for the neuroscience community.

The EEG signals have been modeled using many different methodologies in the literature. An EEG model in terms of nonlinear stochastic differential equation (SDE) could be sufficiently flexible in that it usually contains many parameters, whose values can be tuned to match the model’s output with actual EEG signals for different neurophysiological conditions, such as EC and EO, of CTL subjects or AD patients. Moreover, an SDE model of EEG can be solved by a number of numerical techniques to generate simulated EEG signals superior to actual EEG signals in terms of much higher temporal resolution and much larger number of sample paths available. These are the two main reasons why we choose to work with SDE models of EEG signals. Specifically, we employed a model of stochastic coupled Duffing–van der Pol oscillators proposed by Ref. [1], which is flexible enough to represent the EC and EO conditions for both CTL and AD subjects and straightforward enough to be simulated by using typical numerical techniques for solving SDE. Moreover, the model parameters reported in Ref. [1] were fine tuned against real-world experimental EEG signals of CTL and AD patients with both EC and EO conditions, and therefore, quantitative investigations on the model’s output of simulated signals are sufficiently representative for a large population of healthy and AD subjects.

## 2. Methods

### 2.1. Stochastic Nonlinear Oscillator Models of EEG Signals

A phenomenological model of the EEG based on a coupled system of Duffing–van der Pol oscillators subject to white noise excitation has been introduced [1] with the following form: (1)$$\left\{ \begin{array}{c} {\ddot{x}}_{1}+\left(k_{1}+k_{2}\right)x_{1}-k_{2}x_{2}=-b_{1}x_{1}^{3}-b_{2}{\left(x_{1}-x_{2}\right)}^{3}+\;\epsilon_{1}{\dot{x}}_{1}\left(1-x_{1}^{2}\right),\\ {\ddot{x}}_{2}-k_{2}x_{1}+k_{2}x_{2}=b_{2}{\left(x_{1}-x_{2}\right)}^{3}+\;\epsilon_{2}{\dot{x}}_{2}\left(1-x_{2}^{2}\right)+\mu \;dW, \end{array}\right.$$
where $x_{i},{\dot{x}}_{i},{\ddot{x}}_{i},i=1,2$ are positions, velocities, and accelerations of the two oscillators, respectively. Parameters $k_{i},b_{i},\epsilon_{i},i=1,2$ are the linear stiffness, cubic stiffness, and van der Pol damping coefficient of the two oscillators, respectively. Parameter μ represents the intensity of white noise and dW is a Wiener process representing the additive noise in the stochastic differential system. The physical meanings of these variables and parameters were nicely explained in a schematic figure in Ref. [1].

By using actual EEG signals, Ref. [1] utilized a combination of several different statistical and optimization techniques to fine tune the parameters in the model equations for eyes-closed (EC) and eyes-open (EO) conditions of both healthy control (CTL) subjects and Alzheimer’s disease (AD) patients, and these parameter values for different conditions are summarized in Table 1 and Table 2.

The model Equation (Equation 1) can be easily rewritten in a more standard form of stochastic differential equation (SDE) as follows: (2)$$\left\{ \begin{array}{c} {\dot{x}}_{1}=x_{3},\\ {\dot{x}}_{2}=x_{4},\\ {\dot{x}}_{3}=-\left(k_{1}+k_{2}\right)x_{1}+k_{2}x_{2}-b_{1}x_{1}^{3}-b_{2}{\left(x_{1}-x_{2}\right)}^{3}+\epsilon_{1}x_{3}\left(1-x_{1}^{2}\right),\\ {\dot{x}}_{4}=k_{2}x_{1}-k_{2}x_{2}+b_{2}{\left(x_{1}-x_{2}\right)}^{3}+\epsilon_{2}x_{4}\left(1-x_{2}^{2}\right)+\mu \;dW, \end{array}\right.$$
which is more readily suitable for stochastic simulations.

### 2.2. Initial Conditions (ICs) and Specifications of Stochastic Simulations

For simplicity, we employ the Euler–Maruyama scheme [29] to simulate $2\times 10^{7}$ trajectories in total of the model Equation (Equation 2); although, other more sophisticated methods for stochastic simulations exist. We simulate such a large number of trajectories, because calculations of information geometry theoretic measures rely on accurate estimation of probability density functions (PDFs) of the model’s variables $x_{i}\left(t\right)$, which requires a large number of data samples of $x_{i}\left(t\right)$ at any given time *t*.

Since nonlinear oscillators’ solution is very sensitive to initial conditions, we start the simulation with a certain initial probability distribution (e.g., a Gaussian distribution) for all $x_{1}\left(0\right),x_{2}\left(0\right),x_{3}\left(0\right),x_{4}\left(0\right)$, which means that the 20 million $x_{i}\left(0\right)$(∀i=1,2,3,4) are randomly drawn from a probability density function (PDF) of the initial distribution. The time-step size dt is set to $10^{-6}$ to compensate for the very-high values of stiffness parameters $k_{1}$ and $k_{2}$ in Table 1 and Table 2. The total number of simulation time steps is $1\times 10^{7}$, making the total time range of simulation [0,10]. The $\Delta t=10^{-4}$ is the time interval when the probability density functions (PDFs) $p(x_{1},t)$ and $p(x_{2},t)$ are estimated for calculating information geometry theoretic measures such as information rates and causal information rates, as explained in Section 2.3.

For nonlinear oscillators, different initial conditions can result in dramatically different long-term time evolution. So in order to explore more diverse initial conditions, we simulated the SDE with 6 different initial Gaussian distributions with different means and standard deviations, i.e., $x_{1}\left(0\right)∼N\left(\mu_{x_{1}\left(0\right)},\sigma^{2}\right)$, $x_{2}\left(0\right)∼N\left(\mu_{x_{2}\left(0\right)},\sigma^{2}\right)$, $x_{3}\left(0\right)∼N\left(\mu_{x_{3}\left(0\right)},\sigma^{2}\right)$, $x_{4}\left(0\right)∼N\left(\mu_{x_{4}\left(0\right)},\sigma^{2}\right)$, where the parameters are summarized alongside other specifications in Table 3.

For brevity, in this paper, we use the word “initial conditions” or its abbreviation “IC” to refer to the (set of 4) initial Gaussian distributions from which the 20 million $x_{i}\left(0\right)$(∀i=1,2,3,4) are randomly drawn. For example, the “IC No.6” in Table 3 (and simply “IC6” elsewhere in this paper) refers to the 6th (set of 4) Gaussian distributions with which we start the simulation, and the specifications of this stimulation are listed in the last column of Table 3.

### 2.3. Information Geometry Theoretic Measures: Information Rate and Causal Information Rate

When a stochastic differential equation (SDE) model exhibits non-stationary time-varying effects, nonlinearity, and/or non-Gaussian stochasticity, while we are interested in large fluctuations and extreme events in the solutions, simple statistics such as mean and variance might not suffice to compare the solutions of different SDE models (or same model with different parameters). In such cases, quantifying and comparing the time evolution of probability density functions (PDFs) of solutions will provide us with more information [30]. The time evolution of PDFs can be studied and compared through the framework of information geometry [31], wherein PDFs are considered as points on a Riemannian manifold (which is called the statistical manifold), and their time evolution can be considered as a motion on this manifold. Several different metrics can be defined on a probability space to equip it with a manifold structure, including a metric related to the Fisher Information [32], known as the Fisher Information metric [33,34], which we use in this work:(3)$$g_{\mu \nu }\left(\theta \right)\overset{\mathrm{def}}{=}\int_{X}\frac{\partial \log p(x;\{\theta \})}{\partial \theta_{\mu }}\frac{\partial \log p(x;\{\theta \})}{\partial \theta_{\nu }}p\left(x;\left\{\theta \right\}\right)\;dx.$$
Here, p(x;{θ}) denotes a continuous family of PDFs parameterized by parameters {θ}. If a time-dependent PDF p(x,t) is considered as a continuous family of PDFs parameterized by a single parameter time *t*, the metric tensor is then reduced to a scalar metric:(4)$$g\left(t\right)=\int dx\;\frac{1}{p(x,t)}{\left[\frac{\partial p(x,t)}{\partial t}\right]}^{2}.$$
The infinitesimal distance dL on the manifold is then given by $dL^{2}=g\left(t\right)\;dt^{2}$, where L is called the Information Length and defined as follows:(5)$$L\left(t\right)\overset{\mathrm{def}}{=}\int_{0}^{t}dt_{1}\;\sqrt{\int dx\;\frac{1}{p(x,t_{1})}{\left[\frac{\partial p(x,t_{1})}{\partial t_{1}}\right]}^{2}}.$$
The Information Length L represents the dimensionless distance, which measures the total distance traveled on the statistical manifold. The time derivative of L then represents the speed of motion on this manifold:(6)$$\Gamma \left(t\right)\overset{\mathrm{def}}{=}\lim_{dt\to 0}\frac{dL(t)}{dt}=\sqrt{\int dx\;\frac{1}{p(x,t)}{\left[\frac{\partial p(x,t)}{\partial t}\right]}^{2}},$$
which is referred to as the Information Rate. If multiple variables are involved, such as $x_{i}\left(t\right)$ where i=1,2,3,4 as in the stochastic nonlinear oscillator model Equation (Equation 2), we will use subscript in Γ(t), e.g., $\Gamma_{x_{2}}\left(t\right)$ to denote the information rate of signal $x_{2}\left(t\right)$.

The notion of Causal Information Rate was introduced in Ref. [25] to quantify how one signal instantaneously influences another signal’s information rate. As an example, the causal information rate of signal $x_{1}\left(t\right)$ influencing signal $x_{2}\left(t\right)$’s information rate is denoted and defined by $\Gamma_{x_{1}\to x_{2}}\left(t\right)\overset{\mathrm{def}}{=}\Gamma_{x_{2}}^{*}\left(t\right)-\Gamma_{x_{2}}\left(t\right)$, where
(7)$$\Gamma_{x_{2}}{\left(t\right)}^{2}=\int dx_{2}\;p\left(x_{2},t\right){\left[\partial_{t}\ln p\left(x_{2},t\right)\right]}^{2},$$
and
(8)$$\begin{array}{cc} \Gamma_{x_{2}}^{*}{\left(t\right)}^{2} & \overset{\mathrm{def}}{=}\lim_{t_{*}\to t^{+}}\int dx_{1}\;dx_{2}\;p\left(x_{2},t_{*};x_{1},t\right){\left[\partial_{t_{*}}\ln p\left(x_{2},t_{*}|x_{1},t\right)\right]}^{2}\\ \; & =\lim_{t_{*}\to t^{+}}\int dx_{1}\;dx_{2}\;p\left(x_{2},t_{*};x_{1},t\right){\left[\partial_{t_{*}}\ln p\left(x_{2},t_{*};x_{1},t\right)\right]}^{2}, \end{array}$$
where the relation between conditional, joint, and marginal PDFs $p\left(x_{2},t_{*}|x_{1},t\right)=\frac{p(x_{2},t_{*};x_{1},t)}{p(x_{1},t)}$ and the fact $\partial_{t_{*}}p\left(x_{1},t\right)=0$ for $t_{*}\neq t$ are used in the 2nd equal sign above. $\Gamma_{x_{2}}^{*}$ denotes the (auto) contribution to the information rate from $x_{2}$ itself, while $x_{1}$ is given/known and frozen in time. In other words, $\Gamma_{x_{2}}^{*}$ represents the information rate of $x_{2}$ when the additional information of $x_{1}$ (at the same time with $x_{2}$) becomes available or known. Subtracting $\Gamma_{x_{2}}$ from $\Gamma_{x_{2}}^{*}$ following the definition of $\Gamma_{x_{1}\to x_{2}}$ then gives us the contribution of (knowing the additional information of) $x_{1}$ to $\Gamma_{x_{2}}$, signifying how $x_{1}$ instantaneously influences the information rate of $x_{2}$. One can easily verify that if signals $x_{1}\left(t\right)$ and $x_{2}\left(t\right)$ are statistically independent such that the equal-time joint PDF can be separated as $p\left(x_{1},t;x_{2},t\right)=p\left(x_{1},t\right)\cdot p\left(x_{2},t\right)$, then $\Gamma_{x_{2}}^{*}\left(t\right)$ will reduce to $\Gamma_{x_{2}}\left(t\right)$, making the causal information rate $\Gamma_{x_{1}\to x_{2}}=0$, which is consistent with the assumption that $x_{1}\left(t\right)$ and $x_{2}\left(t\right)$ are statistically independent at the same time *t*.

For numerical estimation purposes, one can derive simplified equations $\Gamma_{x_{2}}{\left(t\right)}^{2}$ = $4\int dx_{2}\;{\left[\partial_{t}\sqrt{p(x_{2},t)}\right]}^{2}$ and $\Gamma_{x_{2}}^{*}{\left(t\right)}^{2}=4\lim_{t_{*}\to t^{+}}\int dx_{1}\;dx_{2}\;{\left[\partial_{t_{*}}\sqrt{p(x_{2},t_{*};x_{1},t)}\right]}^{2}$ to ease the numerical calculations and avoid numerical errors in PDFs (due to finite sample-size estimations using a histogram-based approach) being doubled or enlarged when approximating the integrals in the original Equations (Equation 7) and (Equation 8) by finite summation. On the other hand, the time derivatives of the square root of PDFs are approximated by using temporally adjacent PDFs with each pair of two adjacent PDFs being separated by $\Delta t=10^{-4}$ in time, as mentioned at the end of Section 2.2.

### 2.4. Shannon Differential Entropy and Transfer Entropy

As a comparison with more traditional and established information-theoretic measures, we also calculate differential entropy and transfer entropy using the numerically estimated PDFs and compare them with information rate and causal information rate, respectively.

The Shannon differential entropy of a signal x(t) is defined to extend the idea of Shannon discrete entropy as
(9)$$h\left(x\left(t\right)\right)=E\left[-\ln p\left(x,t\right)\right]=-\int p\left(x,t\right)\ln p\left(x,t\right)\;dx=-\int P\left(dx\left(t\right)\right)\;\ln\frac{P(dx(t))}{\mu (dx)},$$
where μ(dx)=dx is the Lebesgue measure, and P(dx(t))=p(x,t)μ(dx)=p(x,t)dx is the probability measure. In other words, differential entropy is the negative relative entropy (Kullback-Leibler divergence) from the Lebesgue measure (considered as an unnormalized probability measure) to a probability measure *P* (with density *p*). In contrast, information rate $\Gamma_{x}\left(t\right)$ = $\sqrt{\int dx\;p\left(x,t\right){\left[\partial_{t}\ln p\left(x,t\right)\right]}^{2}}=\sqrt{\lim_{dt\to 0}\frac{2}{dt^{2}}\int dx\;p\left(x,t+dt\right)\ln\frac{p(x,t+dt)}{p(x,t)}}$ (see Refs. [24,25,26] for detailed derivations) is related to the rate of change in relative entropy of two infinitesimally close PDFs p(x,t) and p(x,t+dt). Therefore, although differential entropy can measure the complexity of a signal x(t) at time *t*, it neglects how the signal’s PDF p(x,t) changes instantaneously at that time, which is crucial to quantify how new information can be reflected from the instantaneous entropy production rate of the signal x(t). This is the theoretical reason why the information rate is a much better and more appropriate measure than differential entropy for characterizing the neural information processing from EEG signals of the brain, and the practical reason for this will be illustrated in terms of numerical results and discussed at the end of Section 3.3.2 and Section 3.3.3.

The transfer entropy (TE) measures the directional flow or transfer of information between two (discrete-time) stochastic processes. The transfer entropy from a signal $x_{1}\left(t\right)$ to another signal $x_{2}\left(t\right)$ is the amount of uncertainty reduced in future values of $x_{2}\left(t\right)$ by knowing the past values of $x_{1}\left(t\right)$ given past values of $x_{2}\left(t\right)$. Specifically, if the amount of information is measured using Shannon’s (discrete) entropy $H(X_{t})$ = $-\sum_{x}p\left(x,t\right)\log_{2}p\left(x,t\right)$ of a stochastic process $X_{t}$ and conditional entropy $H\left(Y_{t_{2}}|X_{t_{1}}\right)=-\sum_{x,y}p\left(x,t_{1};y,t_{2}\right)\log_{2}p\left(y,t_{2}|x,t_{1}\right)$, the transfer entropy from a process $X_{t}$ to another process $Y_{t}$ (for discrete-time t∈Z) can be written as follows:(10)TEXt→Yt(t)=HYt+1∣Yt:t−(k−1)−HYt+1∣Yt:t−(k−1),Xt:t−(l−1),=−∑yp(yt+1,yt:t−(k−1))log2p(yt+1|yt:t−(k−1))+∑x,yp(yt+1,yt:t−(k−1),xt:t−(l−1))log2p(yt+1|yt:t−(k−1),xt:t−(l−1)),=∑x,yp(yt+1,yt:t−(k−1),xt:t−(l−1))log2p(yt+1|yt:t−(k−1),xt:t−(l−1))p(yt+1|yt:t−(k−1)),(11)=∑x,yp(yt+1,yt:t−(k−1),xt:t−(l−1))log2p(yt+1,yt:t−(k−1),xt:t−(l−1))p(yt:t−(k−1))p(yt:t−(k−1),xt:t−(l−1))p(yt+1,yt:t−(k−1)),
which quantifies the amount of reduced uncertainty in future value $Y_{t+1}$ by knowing the past *l* values of $X_{t}$ given past *k* values of $Y_{t}$, where $Y_{t:t-(k-1)}$ and $X_{t:t-(l-1)}$ are shorthands of past *k* values $Y_{t},Y_{t-1},\ldots ,Y_{t-(k-1)}$ and past *l* values $X_{t},X_{t-1},\ldots ,X_{t-(l-1)}$, respectively.

In order to properly compare with causal information rate signifying how one signal instantaneously influences another signal’s information rate (at the same/equal-time *t*), we set k=l=1 in calculating the transfer entropy between two signals. Also, since the causal information rate involves partial time derivatives, which have to be numerically estimated using temporally adjacent PDFs separated by $\Delta t=10^{-4}$ in time (as mentioned at the end of Section 2.2), the discrete-time t∈Z in transfer entropy should be changed to nΔt with n∈Z. Therefore, the transfer entropy appropriate for comparing with the causal information rate should be rewritten as follows:(12)TEx1→x2(t)=Hx2(t+Δt)∣x2(t)−Hx2(t+Δt)∣x2(t),x1(t),=∑x1,x2p(x2,t+Δt;x2,t;x1,t)log2p(x2,t+Δt|x2,t;x1,t)p(x2,t+Δt|x2,t),(13)=∑x1,x2p(x2,t+Δt;x2,t;x1,t)log2p(x2,t+Δt;x2,t;x1,t)p(x2,t)p(x2,t;x1,t)p(x2,t+Δt;x2,t).
Numerical estimations of the information rate, causal information rate, differential entropy, and transfer entropy are all based on numerical estimation of PDFs using histograms. In particular, in order to sensibly and consistently estimate the causal information rate (e.g., to avoid getting negative values), special caution is required when choosing the binning for histogram estimation of PDFs in calculating $\Gamma_{x_{2}}{\left(t\right)}^{2}=4\int dx_{2}\;{\left[\partial_{t}\sqrt{p(x_{2},t)}\right]}^{2}$ and $\Gamma_{x_{2}}^{*}{\left(t\right)}^{2}=4\lim_{t_{*}\to t^{+}}\int dx_{1}\;dx_{2}\;{\left[\partial_{t_{*}}\sqrt{p(x_{2},t_{*};x_{1},t)}\right]}^{2}$. The finer details for these numerical estimation techniques are elaborated in Appendix A.

## 3. Results

We performed simulations with six different Gaussian initial distributions (with different means and standard deviations summarized in Table 3). Initial Conditions No.1 (IC No.1, or simply IC1) through No.3 (IC3) are Gaussian distributions with a narrow width or smaller standard deviation, whereas IC4 through IC6 have a larger width/standard deviation, and therefore, the simulation results of IC4 through IC6 exhibit more diverse time evolution behaviors (e.g., more complex attractors, as explained next), and hence, the corresponding calculation results are more robust or insensitive to the specific mean values $\mu_{x_{i}\left(0\right)}$’s of the initial Gaussian distributions (see Table 3 for more details). Therefore, in the main text here, we focus on these results from initial Gaussian distributions with wider width/larger standard deviation, and we list complete results from all six initial Gaussian distributions in the Appendix B. Specifically, we found that the results from IC4 through IC6 are qualitatively the same or very similar, and therefore, in the main text here, we illustrate and discuss the results from Initial Conditions No.4 (IC4), which is sufficiently representative for IC5 and IC6, and refer to other IC’s (by referencing the relevant sections in Appendix B or explicitly illustrating the results) if needed.

### 3.1. Sample Trajectories of $X_{1}\left(T\right)$ and $X_{2}\left(T\right)$

To give a basic idea of how the simulated trajectories evolve in time, we start by illustrating 50 sample trajectories of $x_{1}\left(t\right)$ and $x_{2}\left(t\right)$ from the total $2\times 10^{7}$ simulated trajectories for both CTL subjects and AD patients with both EC and EO conditions, which are visualized in Figure 1 and Figure 2. Notice that from Figure 1c, one can see that it takes some time for the trajectories of $x_{2}\left(t\right)$ to settle down on some complex attractors for EC, which suggests a longer memory associated with EC of CTL. This is more evident as shown in the time evolution of PDF $p(x_{2},t)$ in Figure 3c below.

### 3.2. Time Evolution of PDF $P(X_{1},T)$ and $P(X_{2},T)$

The empirical PDFs $p(x_{1},t)$ and $p(x_{2},t)$ can better illustrate the overall time evolution of a large number of trajectories, and they serve as a basis for calculations of information geometry theoretic measures such as information rates and causal information rates. These empirical PDFs are estimated using a histogram-based approach with Rice’s rule [35,36], where the number of bins is $n_{\mathrm{bins}}=2\sqrt[{3}]{n_{\mathrm{samples}}}$, and since we simulated $2\times 10^{7}$ sample trajectories in total, the $n_{\mathrm{bins}}$ is rounded to 542. The centers of bins are plotted on the *y*-axis in sub-figures of Figure 3 and Figure 4, where the function values of $p(x_{1},t)$ and $p(x_{2},t)$ are color-coded following the color bars.

As mentioned in the previous section, from Figure 3c, one can see more clearly that after around t≥5, the trajectories settle down on some complex attractors, and the time evolution of $p(x_{2},t)$ undergoes only minor changes. Meanwhile, from Figure 3a, one can observe that a similar settling down of $x_{1}\left(t\right)$ on some complex attractors happens after around t≥7.5. Therefore, we select only PDFs with t≥7.5 for statistical analysis of information rates and causal information rates to investigate the stationary properties.

From Figure 3 and Figure 4, one can already observe some qualitative differences between healthy control (CTL) subjects and AD patients. For example, the time evolution patterns of $p(x_{1},t)$ and $p(x_{2},t)$ are significantly different when CTL subjects open their eyes from eyes-closed (EC) state, whereas for AD patients, these differences are relatively minor. One of the best ways to provide quantitative descriptions of these differences (instead of being limited to qualitative descriptions) is using information geometry theoretic measures such as information rates and causal information rates, whose results are listed in Section 3.3 and Section 3.4, respectively.

As can be seen from Section B.2, IC1 through IC3 exhibit much simpler attractors than IC4 through IC6. Since the width/standard deviation of the initial Gaussian distributions of IC1 through IC3 is much smaller, they are more sensitive to the specific mean values $\mu_{x_{i}\left(0\right)}$’s of the initial Gaussian distribution, and one can see that IC3’s time evolution behaviors of $p(x_{i},t)$ are somewhat qualitatively different from IC1 and IC2, whereas $p(x_{i},t)$’s time evolution behaviors of IC4 through IC6 are all qualitatively the same.

### 3.3. Information Rates $\Gamma_{x_{1}}\left(t\right)$ and $\Gamma_{x_{2}}\left(t\right)$

Intuitively speaking, the information rate is the (instantaneous) speed of PDF’s motion on the statistical manifold, as each given PDF corresponds to a point on that manifold, and when the time changes, a time-dependent PDF will typically move on a curve on the statistical manifold, whereas a stationary or equilibrium state PDF will remain at the same point on the manifold. Therefore, the information rate is a natural tool to investigate the time evolution of PDF.

Moreover, since the information rate is quantifying instantaneous rate of change in the infinitesimal relative entropy between two adjacent PDFs, it is hypothetically a reflection of neural information processing in the brain, and hence, it may provide important insight into the neural activities in different regions of the brain, as long as the regional EEG signals can be sufficiently collected for calculating the information rates.

#### 3.3.1. Time Evolution

The time evolution of information rates $\Gamma_{x_{1}}\left(t\right)$ and $\Gamma_{x_{2}}\left(t\right)$ are shown in Figure 5a,b for CTL subjects and AD patients, respectively. Since $\Gamma_{x_{1}}\left(t\right)$ and $\Gamma_{x_{2}}\left(t\right)$ quantify the (infinitesimal) relative entropy production rate instantaneously at time *t*, they represent the information-theoretic complexities of signals $x_{1}\left(t\right)$ and $x_{2}\left(t\right)$ of the coupled oscillators, respectively, and are hypothetical reflections of neural information processing in the corresponding regions in the brain.

For example, in Figure 5a, there is a clear distinction between eyes-closed (EC) and eyes-open (EO) for CTL subjects: both $\Gamma_{x_{1}}\left(t\right)$ and $\Gamma_{x_{2}}\left(t\right)$ decrease significantly when healthy subjects open their eyes, which may be interpreted as the neural information processing activities of the corresponding brain regions being “suppressed” by the incoming visual information when eyes are opened from being closed.

Interestingly, when AD patients open their eyes, both $\Gamma_{x_{1}}\left(t\right)$ and $\Gamma_{x_{2}}\left(t\right)$ are increasing instead of decreasing, as shown in Figure 5b. This might be interpreted as that the incoming visual information received when eyes are opened is in fact “stimulating” the neural information processing activities of the corresponding brain regions, which might be impaired or damaged by the relevant mechanism of Alzheimer’s disease (AD).

In Figure 5a,b, we annotate the mean and standard deviation for $\Gamma_{x_{1}}\left(t\right)$ and $\Gamma_{x_{2}}\left(t\right)$ after t≥7.5 in the legend, because as mentioned above, the PDFs of this time range reflect longer-term temporal characteristics, and hence, the corresponding $\Gamma_{x_{1}}\left(t\geq 7.5\right)$ and $\Gamma_{x_{2}}\left(t\geq 7.5\right)$ should reflect more reliable and robust features of neural information processing activities of the corresponding brain regions. Therefore, meaningful statistics will require collecting samples of $\Gamma_{x_{1}}\left(t\right)$ and $\Gamma_{x_{2}}\left(t\right)$ in this time range, for which the results are shown in the section below.

#### 3.3.2. Empirical Probability Distribution (for T≥7.5)

The statistics of $\Gamma_{x_{1}}\left(t\geq 7.5\right)$ and $\Gamma_{x_{2}}\left(t\geq 7.5\right)$ can be further and better visualized using empirical probability distributions of them, as shown in Figure 6. Again, we use histogram-based density estimation with Rice’s rule, and since the time interval Δt for estimating PDFs and computing $\Gamma_{x_{1}}\left(t\right)$ and $\Gamma_{x_{2}}\left(t\right)$ is $10^{-4}$ (whereas the time-step size dt for simulating the SDE model is $10^{-6}$), we collected 24,999 samples of both $\Gamma_{x_{1}}\left(t\right)$ and $\Gamma_{x_{2}}\left(t\right)$ for 7.5≤t<10, and hence, the number of bins following Rice’s rule is rounded to 58. Figure 6 confirms the observation in the previous section, while it also better visualizes the sample standard deviation in the shapes of the estimated PDFs, indicating that the PDFs of both $\Gamma_{x_{1}}\left(t\right)$ and $\Gamma_{x_{2}}\left(t\right)$ are narrowed down when healthy subjects open their eyes but are widened when AD patients do so.

As a comparison, we also calculate more traditional/established information-theoretic measure, namely, the Shannon differential entropy $h(x_{1}\left(t\right))$ and $h(x_{2}\left(t\right))$, and estimate their empirical probability distributions in the same manner as we do for information rates, as shown in Figure 7.

One can see that the empirical distributions of differential entropy $h(x_{1}\left(t\right))$ and $h(x_{2}\left(t\right))$ are not able to make clear distinction between EC and EO conditions, especially for AD patients. This may be better summarized in Table 4, comparing the mean and standard deviation values of information rate vs. differential entropy for the four cases. Therefore, the information rate is a superior measure for quantifying the non-stationary time-varying dynamical changes in EEG signals when switching between EC and EO states and is a better and more reliable reflection of neural information processing in the brain.

#### 3.3.3. Phase Portraits (for T≥7.5)

In addition to empirical statistics of information rates for t≥7.5 in terms of estimated probability distributions, one can also visualize the temporal dynamical features of $\Gamma_{x_{1}}\left(t\right)$ and $\Gamma_{x_{2}}\left(t\right)$ combined using phase portraits, as shown in Figure 8. Notice that when healthy subjects open their eyes, the fluctuation ranges of $\Gamma_{x_{1}}\left(t\right)$ and $\Gamma_{x_{2}}\left(t\right)$ shrink by roughly 5-fold, whereas when AD patients open their eyes, the fluctuation ranges are enlarged.

Moreover, when plotting EC and EO of healthy subjects separately in Figure 9a to zoom into the ranges of $\Gamma_{x_{1}}\left(t\right)$ and $\Gamma_{x_{2}}\left(t\right)$ for EO, one can also see that the phase portrait of EO exhibits a fractal-like pattern, whereas the phase portrait of EC exhibits more regular dynamical features, including an overall trend of fluctuating between bottom left and top right, indicating that the $\Gamma_{x_{1}}\left(t\right)$ and $\Gamma_{x_{2}}\left(t\right)$ are somewhat synchronized, which could be explained by the strong coupling coefficients in Table 1 of healthy subjects. Contrarily, for AD patients, the phase portraits of EC and EO both exhibit fractal-like patterns in Figure 9b.

Same as at the end of Section 3.3.2, as a comparison, we also visualize the phase portraits of Shannon differential entropy $h(x_{1}\left(t\right))$ and $h(x_{2}\left(t\right))$ in Figure A52d and Figure A56 in Section B.4.3, where one can see that it is hard to distinguish the phase portraits of $h(x_{1}\left(t\right))$ vs. $h(x_{2}\left(t\right))$ of AD EC from those of AD EO, as they are qualitatively the same or very similar. Contrarily, in Figure 8, the fluctuation ranges of phase portraits of $\Gamma_{x_{1}}\left(t\right)$ vs. $\Gamma_{x_{2}}\left(t\right)$ are significantly enlarged when AD patients open their eyes. Therefore, this reconfirms our claim at the end of Section 3.3.2 that the information rate is a superior measure than differential entropy in quantifying the dynamical changes in EEG signals when switching between EO and EO states and is a better and more reliable reflection of neural information processing in the brain.

#### 3.3.4. Power Spectra (for T≥7.5)

Another perspective to visualize the dynamical characteristics of $\Gamma_{x_{1}}\left(t\right)$ and $\Gamma_{x_{2}}\left(t\right)$ is by using power spectra, i.e., the absolute values of (fast) Fourier transforms of $\Gamma_{x_{1}}\left(t\right)$ and $\Gamma_{x_{2}}\left(t\right)$, as shown in Figure 10. Frequency-based analyses will not make much sense if the signals or time series of $\Gamma_{x_{1}}\left(t\right)$ and $\Gamma_{x_{2}}\left(t\right)$ have non-stationary time-varying effects, and this is why we only consider time range t≥7.5 for $\Gamma_{x_{1}}\left(t\right)$ and $\Gamma_{x_{2}}\left(t\right)$, when the time evolution patterns of $p(x_{1},t)$ and $p(x_{2},t)$ almost stop changing as shown in Figure 3 (and especially in Figure 3a,c).

The power spectra of $\Gamma_{x_{1}}\left(t\right)$ and $\Gamma_{x_{2}}\left(t\right)$ also exhibit a clear distinction between EC and EO for CTL and AD subjects. Specifically, the power spectra of $\Gamma_{x_{1}}\left(t\right)$ and $\Gamma_{x_{2}}\left(t\right)$ can be fit by power law for frequencies between ∼100 Hz to ∼1000 Hz (the typical sampling frequency of experimental EEG signals is 1000 Hz, whereas most of brain wave’s/neural oscillations’ frequencies are below 100 Hz). From Figure 11a, one can see that power law fit exponents (quantifying how fast the power density decreases with increasing frequency) of $\Gamma_{x_{1}}\left(t\right)$’s and $\Gamma_{x_{2}}\left(t\right)$’s power spectra are largely reduced when healthy subjects open their eyes, which indicates that the strength of noise in $\Gamma_{x_{1}}\left(t\right)$ and $\Gamma_{x_{2}}\left(t\right)$ decreases significantly when switching from EC to EO. Contrarily, for AD patients as shown in Figure 11b, the power law fit exponents of $\Gamma_{x_{1}}\left(t\right)$’s and $\Gamma_{x_{2}}\left(t\right)$’s power spectra increase significantly and slightly, respectively, indicating that the strength of noise in $\Gamma_{x_{1}}\left(t\right)$ and $\Gamma_{x_{2}}\left(t\right)$ increases when switching from EC to EO.

### 3.4. Causal Information Rates $\Gamma_{x_{2}\to X_{1}}\left(T\right),\Gamma_{x_{1}\to X_{2}}\left(T\right)$, and Net Causal Information Rates $\Gamma_{x_{2}\to X_{1}}\left(T\right)-\Gamma_{x_{1}\to X_{2}}\left(T\right)$

The notion of causal information rate was introduced in Ref. [25], which quantifies how one signal instantaneously influences another signal’s information rate. A comparable measure of causality is transfer entropy; however, as shown in Section B.6, our calculation results of transfer entropy are too spiky/noisy to reliably quantify causality, and hence, the results are only included in Appendix as a comparison, which we will discuss at the end of this section. Nevertheless, similar to net transfer entropy, one can calculate the net causal information rate, e.g., $\Gamma_{x_{2}\to x_{1}}\left(t\right)-\Gamma_{x_{1}\to x_{2}}\left(t\right)$, signifying the net causality measure from signal $x_{2}\left(t\right)$ to $x_{1}\left(t\right)$. Since ${\dot{x}}_{2}\left(t\right)$ is the only variable that is directly affected by random noise in the stochastic oscillator model Equation (Equation 2), we calculate $\Gamma_{x_{2}\to x_{1}}\left(t\right)-\Gamma_{x_{1}\to x_{2}}\left(t\right)$ for the net causal information rate of the coupled oscillator’s signal $x_{2}\left(t\right)$ influencing $x_{1}\left(t\right)$.

Notice that for stochastic coupled oscillator model Equation (Equation 2), causal information rates $\Gamma_{x_{2}\to x_{1}}\left(t\right)$ and $\Gamma_{x_{1}\to x_{2}}\left(t\right)$ will reflect how strongly the two oscillators are directionally coupled or causally related. Since signals $x_{1}\left(t\right)$ and $x_{2}\left(t\right)$ are the results of neural activities in the corresponding brain regions, the causal information rates can be used to measure connectivities among different regions of the brain.

#### 3.4.1. Time Evolution

Similar to Section 3.3.1, we also visualize the time evolution of causal information rates $\Gamma_{x_{2}\to x_{1}}\left(t\right)$ and $\Gamma_{x_{1}\to x_{2}}\left(t\right)$ in Figure 12a,b for CTL subjects and AD patients, respectively.

For both CTL and AD subjects, $\Gamma_{x_{2}\to x_{1}}\left(t\right)$ and $\Gamma_{x_{1}\to x_{2}}\left(t\right)$ both decrease when changing from EC to EO, except for AD subjects’ $\Gamma_{x_{2}\to x_{1}}\left(t\right)$ increasing on average. On the other hand, the net causal information rate $\Gamma_{x_{2}\to x_{1}}\left(t\right)-\Gamma_{x_{1}\to x_{2}}\left(t\right)$ changes differently: when healthy subjects open their eyes, it increases and changes from significantly negative on average to slightly positive on average, whereas for AD patients, it increases from almost zero on average to significantly positive on average without net directional change. A possible interpretation might be that, when healthy subjects open their eyes, the brain region generating the signal $x_{2}\left(t\right)$ becomes more sensitive to the noise, causing it to influence $x_{1}\left(t\right)$ more compared to the eyes-closed state.

#### 3.4.2. Empirical Probability Distribution (for T≥7.5)

Similar to Section 3.3.2, we also estimate the empirical probability distributions of $\Gamma_{x_{2}\to x_{1}}\left(t\right)$, $\Gamma_{x_{1}\to x_{2}}\left(t\right)$ and $\Gamma_{x_{2}\to x_{1}}\left(t\right)-\Gamma_{x_{1}\to x_{2}}\left(t\right)$ to better visualize their statistics in Figure 13a. In particular, we plot the empirical probability distributions of $\Gamma_{x_{2}\to x_{1}}\left(t\right)-\Gamma_{x_{1}\to x_{2}}\left(t\right)$ for both healthy and AD subjects with both EC and EO conditions together in Figure 13b, in order to better visualize and compare net causal information rates’ changes when CTL and AD subjects open their eyes. It can be seen that the estimated PDF of $\Gamma_{x_{2}\to x_{1}}\left(t\right)-\Gamma_{x_{1}\to x_{2}}\left(t\right)$ shrinks its width in shape when healthy subjects open their eyes. Combining with the observation that the magnitude of sample mean of $\Gamma_{x_{2}\to x_{1}}\left(t\right)-\Gamma_{x_{1}\to x_{2}}\left(t\right)$ is close to 0 for healthy subjects with eyes opened, a possible interpretation might be that the directional connectivity between brain regions generating signals $x_{1}\left(t\right)$ and $x_{2}\left(t\right)$ is reduced to almost zero, either by incoming visual information received by opened eyes or due to the brain region generating signal $x_{2}\left(t\right)$ becoming more sensitive to noise when eyes are opened. Contrarily, the estimated PDF of $\Gamma_{x_{2}\to x_{1}}\left(t\right)-\Gamma_{x_{1}\to x_{2}}\left(t\right)$ for AD patients qualitatively change in an inverse direction to become widened in shape.

As mentioned earlier, as a comparison, we also calculate more traditional/established information-theoretic measure of causality, i.e., transfer entropy (TE), and estimate their empirical probability distributions in the same manner as we do for causal information rates, as shown in Figure 14.

One can see that the empirical distributions of transfer entropy ${\mathrm{TE}}_{x_{2}\to x_{1}}\left(t\right)$ and ${\mathrm{TE}}_{x_{1}\to x_{2}}\left(t\right)$, as well as net transfer entropy ${\mathrm{TE}}_{x_{2}\to x_{1}}\left(t\right)-{\mathrm{TE}}_{x_{1}\to x_{2}}\left(t\right)$ are not able to make clear distinction between EC and EO conditions, especially for AD patients in terms of net transfer entropy. This may be better summarized in Table 5, comparing the mean and standard deviation values of causal information rate vs. transfer entropy for the four cases.

Moreover, the magnitude of numeric values of transfer entropy and net transfer entropy is ∼$10^{-2}$ or ∼$10^{-3}$, which is too close to zero, making it too noise-like or unreliable to quantify causality. Therefore, the causal information rate is a much superior measure than transfer entropy in quantifying causality, and since causal information rate quantifies how one signal instantaneously influences another signal’s information rate (which is a reflection of neural information processing in corresponding brain region), it can be used to measure directional or causal connectivities among different brain regions.

## 4. Discussion

A major challenge for practical usage of information geometry theoretic measures on real-world experimental EEG signals is that they require a significant amount of data samples to estimate the probability density functions. For example, in this work, we simulated $2\times 10^{7}$ trajectories or sample paths of the stochastic nonlinear coupled oscillator models, such that at any time instance, we always have a sufficient amount of data samples to accurately estimate the time-dependent probability density functions with a histogram-based approach. This is usually not possible for experimental EEG signals which often contain only one trajectory for each channel, and one has to use a sliding window-based approach to collect data samples for histogram-based density estimation. This approach implicitly assumes that the EEG signals are stationary within each sliding time window, and hence, one has to balance between the sliding time window’s length and number of available data samples, in order to account for non-stationarity while still having enough data samples to accurately and meaningfully estimate the time-dependent probability densities. And therefore, this approach will not work very well if the EEG signals exhibit severely non-stationary time-varying effects, requiring a very short length of sliding windows, which will contain too few data samples.

An alternative approach to overcome this issue is using kernel density estimation to estimate the probability density functions, which usually requires a much smaller number of data samples while still being able to approximate the true probability distribution with acceptable accuracy. However, this approach typically involves a very high computational cost, limiting its practical use for many cases such as computational resource-limited scenarios. A proposed method to avoid this is using the Koopman operator theoretic framework [37,38] and its numerical techniques applicable to experimental data in a model-free manner, since the Koopman operator is the left-adjoint of the Perron–Frobenious operator evolving the probability density functions in time. This exploration will be left for our future investigation.

## 5. Conclusions

In this work, we explore information geometry theoretic measures to characterize neural information processing from EEG signals simulated by stochastic nonlinear coupled oscillator models. In particular, we utilize information rates to quantify the time evolution of probability density functions of simulated EEG signals and utilize causal information rates to quantify one signal’s instantaneous influence on another signal’s information rate. The parameters of the stochastic nonlinear coupled oscillator models of EEG were fine tuned for both healthy subjects and AD patients, with both eyes-closed and eyes-open conditions. By using information rates and causal information rates, we find significant and interesting distinctions between healthy subjects and AD patients when they change their eyes’ open/closed status. These distinctions may be further related to differences in neural information processing activities of the corresponding brain regions (for information rates) and to differences in connectivities among these brain regions (for causal information rates).

Compared to more traditional or established information-theoretic measures such as differential entropy and transfer entropy, our results show that information geometry theoretic measures such as information rate and causal information rate are superior to their more traditional counterparts, respectively (information rate vs. differential entropy, and causal information rate vs. transfer entropy). Since information rates and causal information rates can be applied to experimental EEG signals in a model-free manner, and they are capable of quantifying non-stationary time-varying effects, nonlinearity, and non-Gaussian stochasticity presented in real-world EEG signals, we believe that these information geometry theoretic measures can become an important and powerful tool-set for both understanding neural information processing in the brain and diagnosis of neurological disorders such as Alzheimer’s disease in this work.

## Figures and Tables

**Figure 1 entropy-26-00213-f001:**
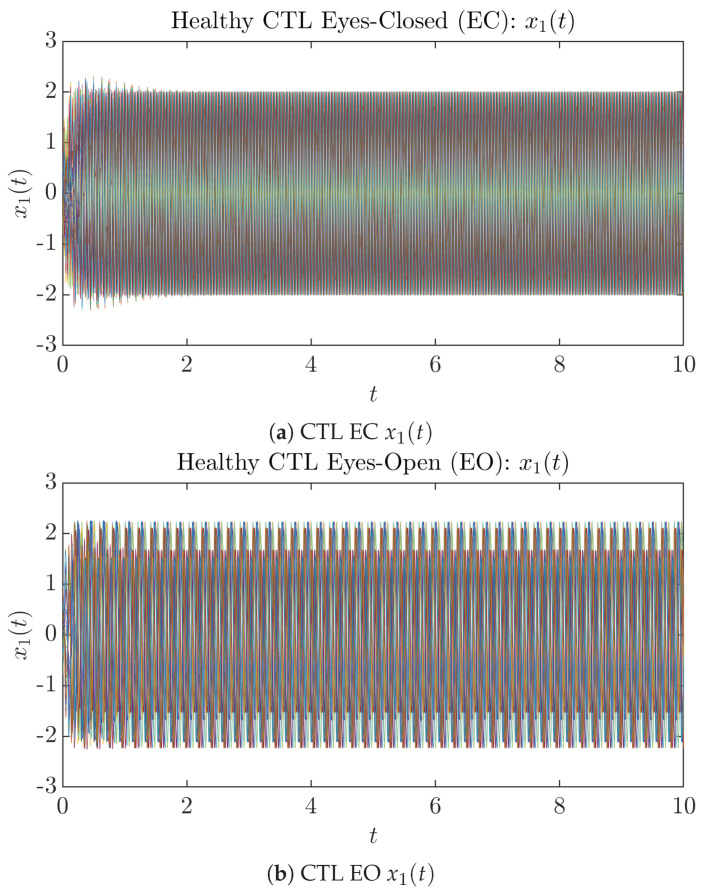
Fifty Sample trajectories of healthy CTL subjects. Each single trajectory is labeled by a different color.

**Figure 2 entropy-26-00213-f002:**
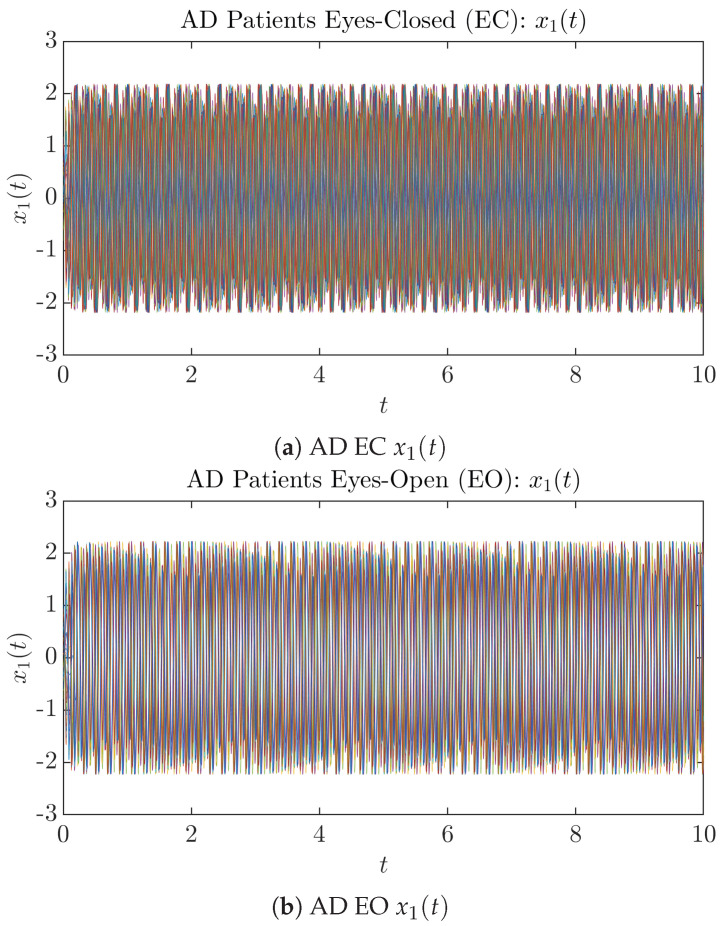
Fifty Sample trajectories of AD patients. Each single trajectory is labeled by a different color.

**Figure 3 entropy-26-00213-f003:**
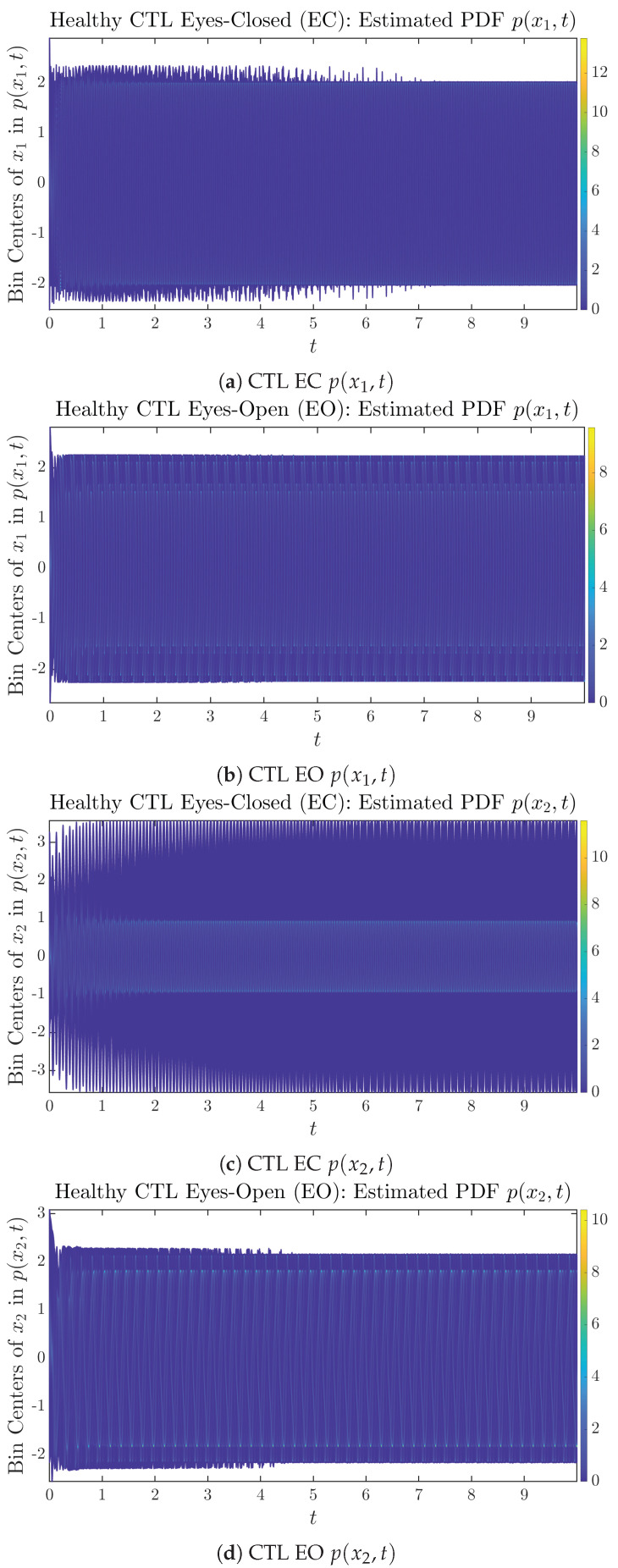
Time evolution of estimated PDFs of healthy CTL subjects.

**Figure 4 entropy-26-00213-f004:**
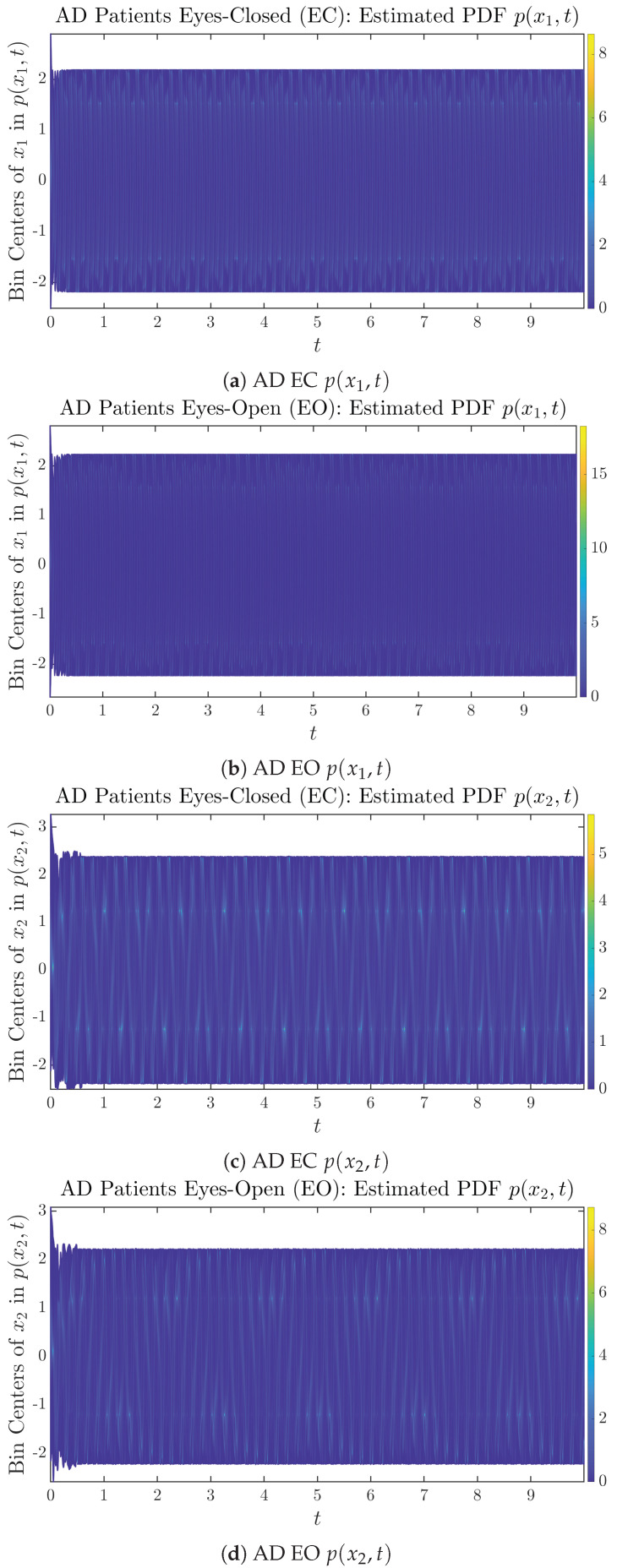
Time evolution of estimated PDFs of AD patients.

**Figure 5 entropy-26-00213-f005:**
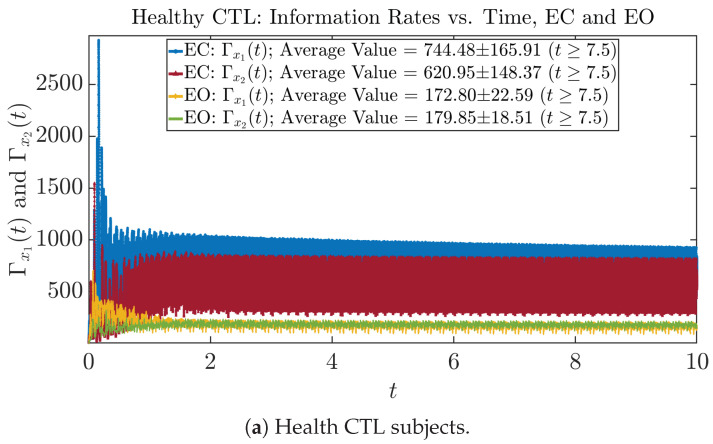
Information rates along time of CTL and AD subjects.

**Figure 6 entropy-26-00213-f006:**
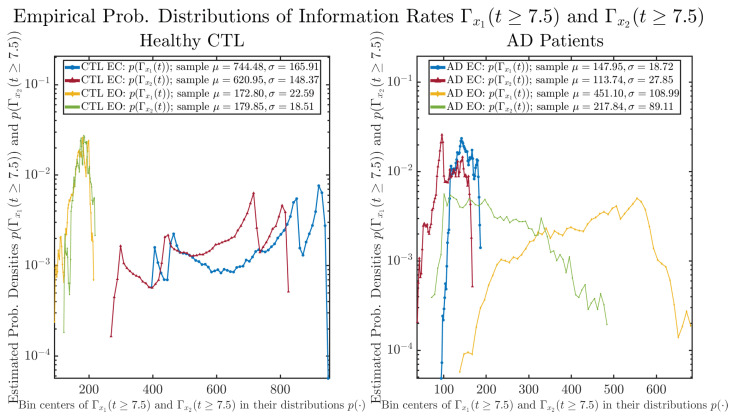
Empirical probability distributions of information rates $\Gamma_{x_{1}}\left(t\right)$ and $\Gamma_{x_{2}}\left(t\right)$(t≥7.5).

**Figure 7 entropy-26-00213-f007:**
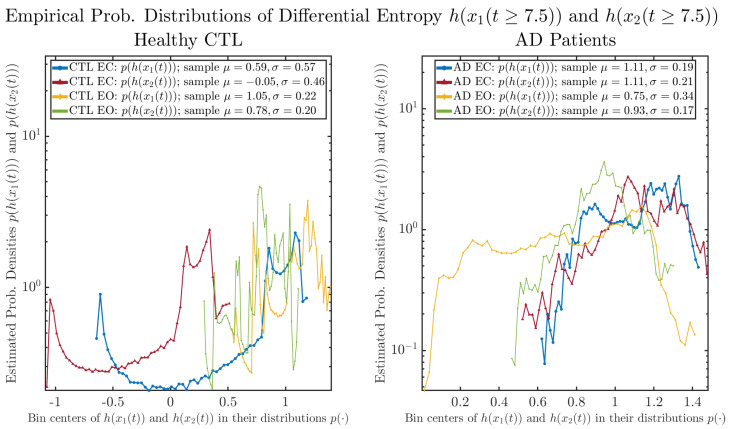
Empirical probability distributions of differential entropy $h(x_{1}\left(t\right))$ and $h(x_{2}\left(t\right))$(t≥7.5).

**Figure 8 entropy-26-00213-f008:**
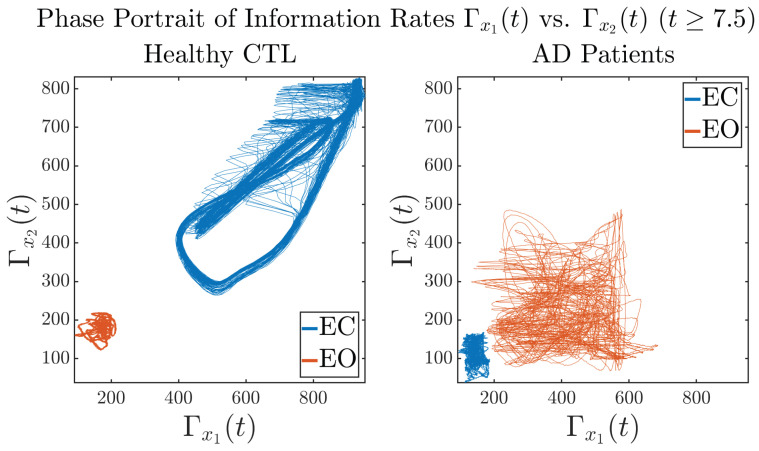
Phase portraits of information rates $\Gamma_{x_{1}}\left(t\right)$ vs. $\Gamma_{x_{2}}\left(t\right)$(t≥7.5).

**Figure 9 entropy-26-00213-f009:**
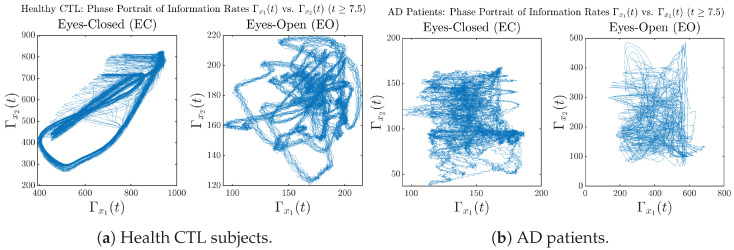
Phase portraits of information rates $\Gamma_{x_{1}}\left(t\right)$ vs. $\Gamma_{x_{2}}\left(t\right)$(t≥7.5) of CTL and AD subjects.

**Figure 10 entropy-26-00213-f010:**
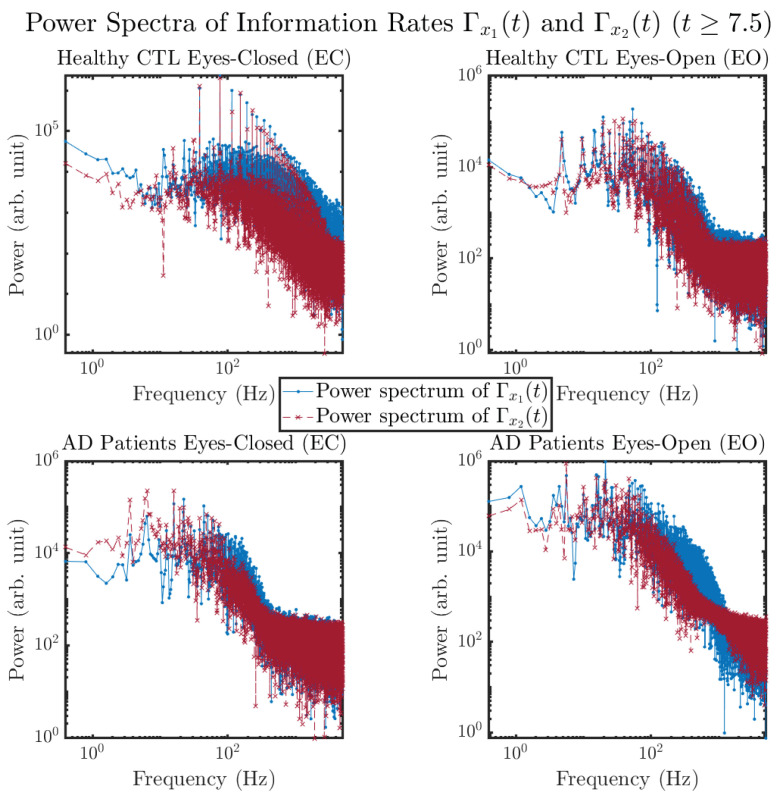
Power spectra of information rates $\Gamma_{x_{1}}\left(t\right)$ and $\Gamma_{x_{2}}\left(t\right)$(t≥7.5).

**Figure 11 entropy-26-00213-f011:**
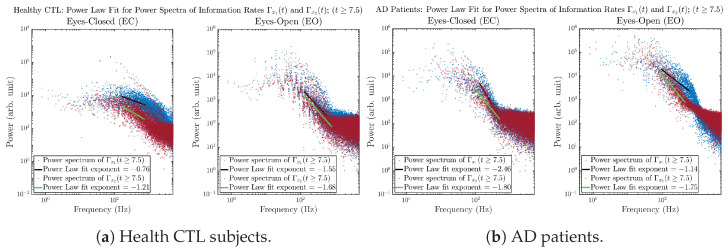
Power law fit for power spectra of information rates $\Gamma_{x_{1}}\left(t\right)$ and $\Gamma_{x_{2}}\left(t\right)$(t≥7.5) of CTL and AD subjects.

**Figure 12 entropy-26-00213-f012:**
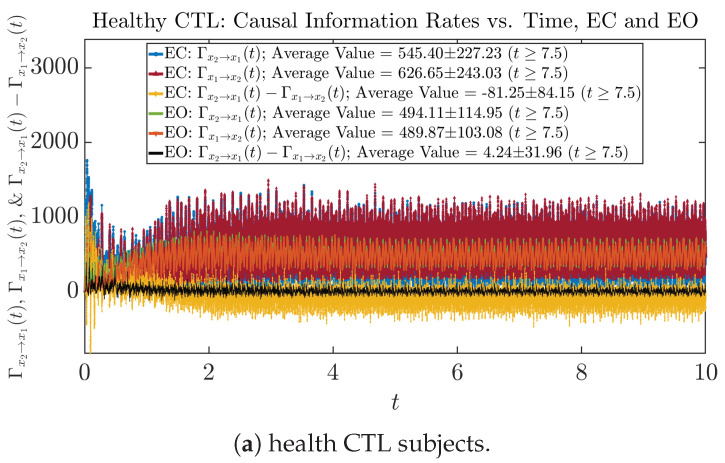
Causal information rates along time of CTL and AD subjects.

**Figure 13 entropy-26-00213-f013:**
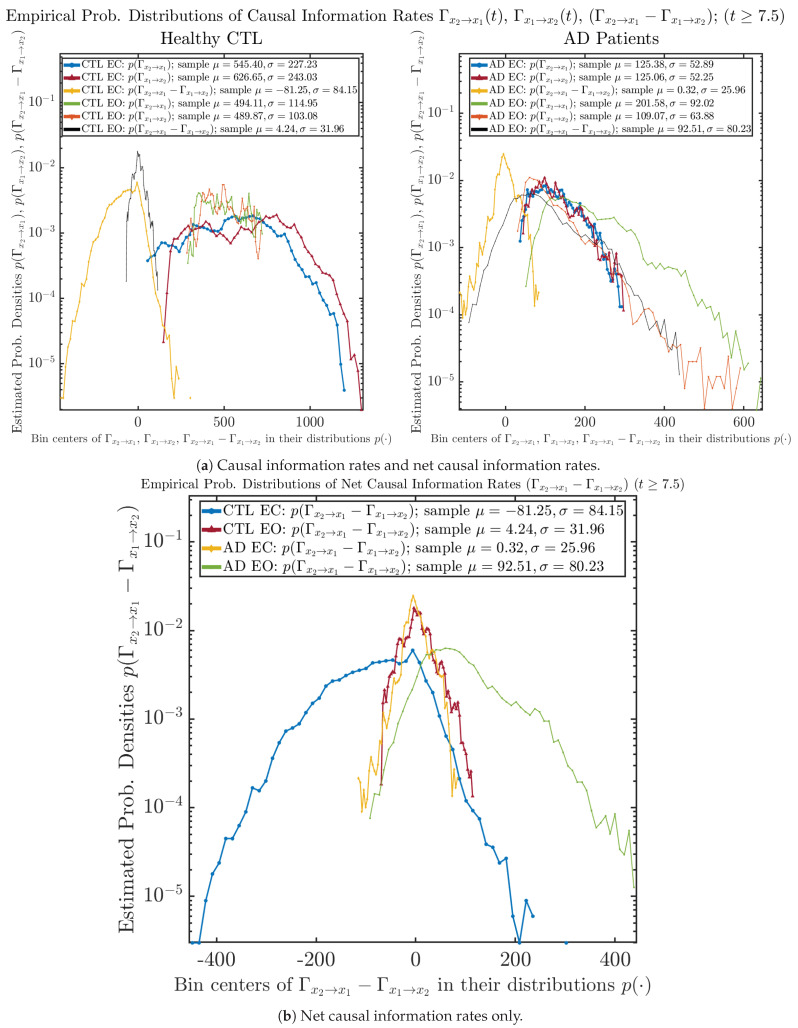
Empirical probability distributions of causal information rates and net causal information rates (t≥7.5).

**Figure 14 entropy-26-00213-f014:**
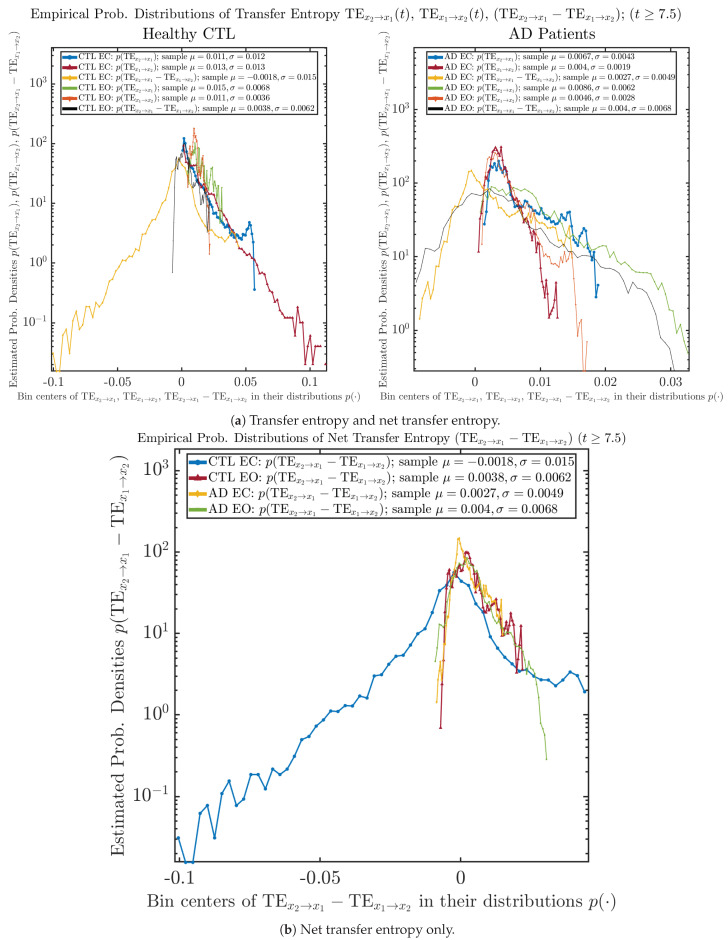
Empirical probability distributions of transfer entropy and net transfer entropy (t≥7.5).

**Table 1 entropy-26-00213-t001:** Optimal parameters of the Duffing–van der Pol oscillator for EC and EO of healthy control (CTL) subjects.

Parameter	Eyes-Closed (EC)	Eyes-Open (EO)
$k_{1}$	7286.5	2427.2
$k_{2}$	4523.5	499.92
$b_{1}$	232.05	95.61
$b_{2}$	10.78	103.36
$\epsilon_{1}$	33.60	48.89
$\epsilon_{2}$	0.97	28.75
μ	2.34	1.82

**Table 2 entropy-26-00213-t002:** Optimal parameters of the Duffing–van der Pol oscillator for EC and EO of Alzheimers disease (AD) patients.

Parameter	Eyes-Closed (EC)	Eyes-Open (EO)
$k_{1}$	1742.1	3139.9
$k_{2}$	1270.8	650.32
$b_{1}$	771.99	101.1
$b_{2}$	1.91	81.3
$\epsilon_{1}$	63.7	56.3
$\epsilon_{2}$	20.7	19.12
μ	1.78	1.74

**Table 3 entropy-26-00213-t003:** Initial conditions (IC): $x_{1}\left(0\right),x_{2}\left(0\right),x_{3}\left(0\right),x_{4}\left(0\right)$ are randomly drawn from Gaussian distributions $N(\mu_{x_{i}\left(0\right)},\sigma^{2})$ with different $\mu_{x_{i}\left(0\right)}$’s and σ’s (i=1,2,3,4).

	IC No.1	IC No.2	IC No.3	IC No.4	IC No.5	IC No.6
$\mu_{x_{1}\left(0\right)}$	1.0	0.9	0.2	0.1	0.5	0.2
$\mu_{x_{2}\left(0\right)}$	0.5	0.1	0.5	0.5	0.9	0.9
$\mu_{x_{3}\left(0\right)}$	0	1.0	0.5	0.2	1.0	0.1
$\mu_{x_{4}\left(0\right)}$	0	0.5	1.0	1.0	0.8	0.5
σ	0.1	0.1	0.1	0.5	0.5	0.5
Num. of trajectories	$2\times 10^{7}$	$2\times 10^{7}$	$2\times 10^{7}$	$2\times 10^{7}$	$2\times 10^{7}$	$2\times 10^{7}$
dt	$10^{-6}$	$10^{-6}$	$10^{-6}$	$10^{-6}$	$10^{-6}$	$10^{-6}$
Δt	$10^{-4}$	$10^{-4}$	$10^{-4}$	$10^{-4}$	$10^{-4}$	$10^{-4}$
Num. of time-steps	$1\times 10^{7}$	$1\times 10^{7}$	$1\times 10^{7}$	$1\times 10^{7}$	$1\times 10^{7}$	$1\times 10^{7}$
Total range of *t*	[0,10]	[0,10]	[0,10]	[0,10]	[0,10]	[0,10]

**Table 4 entropy-26-00213-t004:** Mean and standard deviation values (μ±σ) of information rates $\Gamma_{x_{1}}\left(t\right)$ & $\Gamma_{x_{2}}\left(t\right)$ vs. differential entropy $h(x_{1}\left(t\right))$ & $h(x_{2}\left(t\right))$(t≥7.5).

	CTL EC	CTL EO	AD EC	AD EO
$\Gamma_{x_{1}}\left(t\right)$	744.48 ± 165.91	172.80 ± 22.59	147.95 ± 18.72	451.10 ± 108.99
$\Gamma_{x_{2}}\left(t\right)$	620.95 ± 148.37	179.85 ± 18.51	113.74 ± 27.85	217.84 ± 89.11
$h(x_{1}\left(t\right))$	0.59 ± 0.57	1.05 ± 0.22	1.11 ± 0.19	0.73 ± 0.34
$h(x_{2}\left(t\right))$	−0.05 ± 0.46	0.78 ± 0.20	1.11 ± 0.21	0.93 ± 0.17

**Table 5 entropy-26-00213-t005:** Mean and standard deviation values (μ±σ) of causal information rates $\Gamma_{x_{2}\to x_{1}}\left(t\right)$, $\Gamma_{x_{1}\to x_{2}}\left(t\right)$ and net causal information rates $\Gamma_{x_{2}\to x_{1}}\left(t\right)-\Gamma_{x_{1}\to x_{2}}\left(t\right)$ vs. transfer entropy ${\mathrm{TE}}_{x_{2}\to x_{1}}\left(t\right)$, ${\mathrm{TE}}_{x_{1}\to x_{2}}\left(t\right)$ and net transfer entropy ${\mathrm{TE}}_{x_{2}\to x_{1}}\left(t\right)-{\mathrm{TE}}_{x_{1}\to x_{2}}\left(t\right)$(t≥7.5).

	CTL EC	CTL EO	AD EC	AD EO
$\Gamma_{x_{2}\to x_{1}}\left(t\right)$	545.40 ± 227.23	494.11 ± 114.95	125.38 ± 52.89	201.58 ± 92.02
$\Gamma_{x_{1}\to x_{2}}\left(t\right)$	626.65 ± 243.03	489.87 ± 103.08	125.06 ± 52.25	109.07 ± 63.88
$\Gamma_{x_{2}\to x_{1}}\left(t\right)-\Gamma_{x_{1}\to x_{2}}\left(t\right)$	−81.25 ± 84.15	4.24 ± 31.96	0.32 ± 25.96	92.51 ± 80.23
${\mathrm{TE}}_{x_{2}\to x_{1}}\left(t\right)$	0.011 ± 0.012	0.015 ± 0.0068	0.0067 ± 0.0043	0.0086 ± 0.0062
${\mathrm{TE}}_{x_{1}\to x_{2}}\left(t\right)$	0.013 ± 0.013	0.011 ± 0.0036	0.004 ± 0.0019	0.0046 ± 0.0028
${\mathrm{TE}}_{x_{2}\to x_{1}}\left(t\right)-{\mathrm{TE}}_{x_{1}\to x_{2}}\left(t\right)$	−0.0018 ± 0.015	0.0038 ± 0.0062	0.0027 ± 0.0049	0.004 ± 0.0068

## Data Availability

The stochastic simulation and calculation scripts will be made publicly available in an open repository, which is likely to be updated under https://github.com/jia-chenhua?tab=repositories or https://gitlab.com/jia-chen.hua (both accessed on 17 February 2024).

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
