# Peer review of "Information Geometry Theoretic Measures for Characterizing Neural Information Processing from Simulated EEG Signals"

_entropy, 2024, doi:10.3390/e26030213_

Round 1
Reviewer 1 Report
Comments and Suggestions for Authors
This manuscript describes a new set of metrics that are being proposed to better characterize EEG time series that allows one to capture differential characteristics even in the cases of nonstationary and nonlinear behavior. The manuscript is well written and clear. Overall the results are interesting and convincing based on the figures provided. However, the authors did not provide any quantitative comparison of the information vs. entropy measures. The figures show clear differences between these metrics, and thus these metrics should be able to be quantified. The natural extension to the methodological comparison is to then demonstrate quantitatively that the information metrics quantitatively perform better than entropy measures in differentiating between Ctrl and AD. This information can be incorporated in tables that accompany the figures (but include data for both information and entropy measures in the main manuscript). It is also important to discuss in more detail now this approach could be used in empirical EEG data.
“Researches” is not a word. You can use “numerous research studies” or “numerous studies”
Line 178: “the diagnosis purposes” the is singular, but purposes is plural – please resolve.
Line 184: Consider stating briefly why it matters that these prior approaches didn’t account for nonlinearities, etc. How will it influence interpretation.
Line 231: “of both of” delete 2nd “of”
Line 267: “an stochastic” should be “a stochastic”
Line 451: use “EO” twice rather than EO/EC
In all figures: The legend lines (showing the color of each condition) are very narrow and difficult to see. It would be beneficial to make these more visible.
The two plots in Figure 7 are not on the same scale. It seems appropriate to scale them the same to allow for more direct comparison between Ctrl and AD data.
Figure 8, it is unclear why the patterns of AD and EO of Ctrl are referred to as fractal patterns. These three are clearly less organized than EC of Ctrls, but naming it as a fractal pattern seems a stretch.
Line 475: The comparison is completed based on visual inspection. It would be more useful in the comparison between entropy measures and information measures would be quantified instead. Similar comment for the previous section.
Figure 11. It seems there must be a way to more efficiently summarize the results presented in this figure. Overlapping the evolution makes it exceedingly difficult to definitively see differences that are stated by the authors. Why not include a distribution of values using a violin plot (separate for each condition) to demonstrate the differences for t>7.5 rather than just including the mean in the legend.
Line 574: “characterizing” should be “characterize”
Discussion: The inability to obtain a high number of trajectories from empirical EEG data is a significant challenge. I feel this needs to be addressed more completely. Clearly, the brain never starts again at the same initial conditions (as one can do with simulations), so even multiple measurements will not mimic the situation provided in this manuscript. Thus, it seems reasonable to use segments of EEG data (window) as separate trajectories. What are the likely problems with this in regards to the proposed method – how can the problems be minimized to better guide data collection. The problem for your initial argument is that you said the information metrics can account for nonstationarity and nonlinearity, but in the discussion you say that nonstationarity will interfere with the windowing approach. This needs to be addressed – can this approach account for nonstationarity or not. It seems the purpose of the variability in initial conditions is to address this issue, but the results from the manuscript are not brought into the discussion.
Comments on the Quality of English LanguageOverall the English is very good. Some specific corrections are included in the comments to authors.
Reviewer 2 Report
Comments and Suggestions for Authors
Quantifying the dynamics of the EEG is an important problem. The authors attempt to use a transient that lasts about 0.5 sec (i.e. around 100 samples for fs=200 Hz) between eye close to eye open to quantify different diseases. This is a very difficult task because of the small number of samples involved, and the difficulties of quantifying statistical differences based on pdf descriptors (Fisher information or Shannon entropy).
The idea of approximating the EEG dynamics by constant parameter chaotic oscillators was never credible because the signal produced does not match the EEG dynamics even to the coarse visual analysis….. so the basis of the paper is flawed. There are more biological plausible methodologies to create synthetic EEG but they are not mentioned in the paper (see Walter Freeman “A method for simulating the properties of background “spontaneous” EEG”). I looked at the figures 1,2,3, and the signal there are TOTALLY different from EEG….. this cannot be a good model for EEG.
The paper is very misleading although the title mentions “simulated EEG” because it is not at all related to neural information processing. It is simply a study of the application of information descriptors to segments produced by two synthetic chaotic models with different sets of parameters. The paper format looks like a report, not a paper. There are many figures with different settings that are unrelated with the goals, and it is unclear why they are there……. Anyone knowledgeable of chaotic oscillators knows that initial conditions will create infinitely different trajectories (the attractor). So there is no reason to show 30 of them !!!!. Furthermore, since the dynamical model parameters are different, the attractor will be modified and the value of the descriptors will differ…. But what is the conclusion of all this for the goal of the paper? That dynamic trajectories are different? It is OBVIOUS because the model parameters are different…..
The mathematics are also quite naïve. The definition of differential entropy is quite simple, but the authors write ” In other words, differential entropy is the negative relative entropy (Kullback-Leibler divergence) from the Lebesgue measure (considered as an unnormalized probability measure) to a probability measure P (with density p)”, which does not make any sense.
More importantly the authors never realize that estimating pdfs from dynamic data is a hard problem because in probability theory the samples are required to be IID. They never tested if the data created by the dynamical models was IID, but very likely they are not…. So, nothing is consistent in their numbers and the conclusions are unsupported.
Comments on the Quality of English Language
poor English, but OK
Round 2
Reviewer 1 Report
Comments and Suggestions for Authors
Overall the authors have responded well to the prior critiques.
Minor edit line 597: "fewer" should be "few"